# Conventionalization of Iconic Handshape Preferences in Family Homesign Systems

**Madeline Quam** [1,2,*], **Diane Brentari** [3] **and Marie Coppola** [1,2,4]

1   Department of Psychological Sciences, University of Connecticut, Storrs, CT 06268, USA; marie.coppola@uconn.edu
2   The Connecticut Institute for the Brain and Cognitive Sciences, Storrs, CT 06268, USA
3   Department of Linguistics, University of Chicago, Chicago, IL 60637, USA; dbrentari@uchicago.edu
4   Department Linguistics, University of Connecticut, Storrs, CT 06268, USA
*   Correspondence: madeline.quam@uconn.edu

**Abstract:** Variation in the linguistic use of handshapes exists across sign languages, but it is unclear how these iconic handshape preferences arise and become conventionalized. In order to understand the factors that shape such handshape preferences in the earliest stages of language emergence, we examined communication within family homesign systems. Homesigners are deaf individuals who have not acquired a signed or spoken language and who innovate unique gesture systems to communicate with hearing friends and family ("communication partners"). We analyzed how characteristics of participants and stimulus items influence handshape preferences and conventionalization. Participants included 11 deaf homesigners, 24 hearing communication partners (CPs), and 8 hearing non-signing adults from Nicaragua. Participants were asked to label items using gestures or signs. The handshape type (Handling, Object, or combined Handling+Object) was then coded. The participants and groups showed variability in iconic handshape preferences. Adult homesigners' families demonstrated more conventionalization than did child homesigners' families. Adult homesigners also used a combined Handling + Object form more than other participants. Younger CPs and those with fewer years of experience using a homesign system showed greater conventionalization. Items that elicited a reliable handshape preference were more likely to elicit Handling rather than Object handshapes. These findings suggest that similarity in terms of handshape type varies even within families, including hearing gesturers in the same culture. Although adult homesigners' families were more conventionalized than child homesigners' families, full conventionalization of these handshape preferences do not seem to appear reliably within two to three decades of use in a family when only one deaf homesigner uses it as a primary system.

**Keywords:** homesign; sign language emergence; conventionalization; handshape; iconicity





## 1. Introduction

Since Plato's dialogue *Cratylus*, researchers have been intrigued by the process of naming in spoken and, more recently, in signed languages. Decades of work on sign languages demonstrate that social conventionalization and natural iconic affordances both play important roles, as does the arbitrary nature of linguistic form. Research on "patterned iconicity", a term created by Padden and her colleagues, which refers to the repeated use of iconic strategies for signs within a certain category (Padden et al. 2013, 2015; Hwang et al. 2017), has demonstrated shared preferences for different types of iconicity when naming objects in both sign languages and in the gestures made by hearing people.

Typological variation in handshapes exists across sign languages (Brentari et al. 2015; Eccarius 2008), but it is unclear how iconic handshape preferences arise and become conventionalized. In this paper, we analyze several factors that may be important in tool naming, particularly related to handshape, revisiting this issue in an important population

that naturally engages in the creation of names in their daily lives. The current study investigates the development of iconic handshape preferences by turning to the case of homesigners and their communication partners. Specifically, we examine handshape type preferences for tools within and across individuals and in families with and without a deaf homesigning family member to determine if there is an underlying universal handshape type preference and whether family members who communicate with each other frequently converge on a preference.

### 1.1. Iconic Handshape Preferences in Sign Languages

One way to classify handshape is by iconic class, as either Handling (i.e., the hand represents a hand manipulating an item) or Object (i.e., the hand resembles the item). The Handling/Object distinction is robust and systematically used in a variety of ways, and previous work has shown that handshape preference in sign languages is used both lexically and grammatically. Padden et al. (2015) analyzed the productions of ASL signers and gesturers in the United States and found that both groups used Handling handshapes more frequently to describe actions and Object handshapes more frequently to describe static objects. Hunsicker and Goldin-Meadow (2013) found a child homesigner used handshape class (handling/object) to distinguish nouns and verbs at an early stage of development. The Handling/Object distinction is therefore used to mark a distinction between lexical classes (noun/verb), even in homesign gesture systems in which structured linguistic input is not available.

Within verbs, handshape class (i.e., Handling vs. Object) is used grammatically to mark agent versus no-agent contexts in Nicaraguan Sign Language, American Sign Language, Hong Kong Sign Language, British Sign Language, and Italian Sign Language (Benedicto and Brentari 2004; Goldin-Meadow et al. 2015; Brentari et al. 2015, 2020).

For nouns, the focus of the current study, handshape type is often more uniform within a given language. For example, for lexical items referring to tools, Object handshapes are preferred in American Sign Language (ASL), while Handling handshapes are preferred in New Zealand Sign Language (NZSL) (Padden et al. 2013). San Juan Quiahije Chatino Sign Language (CSL), an emerging sign language, has also demonstrated a Handling preference for tools (Hou 2018). Note that when we use the term emerging sign languages, we are referring to languages that are relatively new (i.e., have existed for decades, rather than centuries or millennia), have a small number of initial users, and may exhibit more variety or higher rates of change (see Le Guen et al. 2020 for a more detailed definition of emerging sign languages). By describing a language as emerging, we are by no means implying any sort of hierarchy amongst languages and want to be clear that we are not suggesting that emerging languages are in any way less than established languages (see Braithwaite 2020 for further discussion). Rather, we are making a distinction between emerging languages and languages with much longer histories, which as a result have different characteristics. Because we take a developmental perspective on language creation and language genesis, we use the term "emerging" in the same spirit as one characterizes the developing language of a child. That is, the language is in flux and, therefore, can reveal the capacities and processes that allow it to emerge, which we propose are the same as those that allow children to acquire the languages around them so effortlessly (Senghas and Coppola 2001; Senghas 2019).

Since we can observe differences across languages for Handling or Object preferences, then presumably, during the emergence of a system, initial users of the system have the opportunity to somehow choose a handshape preference type. As this is likely not a conscious decision, we would like to understand the factors that go into settling on an iconic handshape preference. However, not every sign language uses patterned iconicity as a strategy, for example, the Yucatec Mayan Sign Languages, a group of relatively young village sign languages (Safar and Chan 2020). Since this systematic use of Handling/Object can be used in a variety of morphological and syntactic contrasts, and yet does not seem to be present in every sign language, patterned iconicity may not be a universal phenomenon

early in language emergence, but instead may only become evident later. If there is indeed a universal cognitive bias towards Handling or Object, or if there are inherent properties of the items themselves, we might observe this when a system is emerging (Brentari et al. 2012). In other words, if this type of iconic handshape preference is available early, we may observe it in homesign systems as well, but if it emerges later, we would only see it in established sign languages and not in homesign systems. In order to understand how iconic handshape preferences develop for labeling objects, we must examine cases other than signers of established sign languages, such as homesigners and hearing gesturers.

Differences in handshape preferences between signers and non-signers are also observed; in general, hearing silent gesturers (i.e., hearing individuals with no exposure to a sign language who are asked to label an item or describe an event without speaking) tend to use Handling handshapes (Padden et al. 2015). Additionally, hearing people silently gesturing do not always show the same preferences and patterns as signers from the same community. Even in childhood, signers become attuned to the contrast between Object and Handling Handshapes and in turn use strategies to make those distinctions, something that hearing gesturers do not do (Brentari et al. 2015). There are also differences in the complexity of handshapes between signers and gesturers. In Nicaragua, Italy, and the United States, signers of Italian Sign Language (LIS) and ASL show higher finger complexity in Object Handshapes and higher joint complexity for Handling Handshapes, while hearing Italian and American gesturers show the reverse pattern (Brentari et al. 2012, 2017). Clearly signers and non-signers use iconic handshape preferences differently; specifically, only signers use this handshape preference grammatically. In order to understand how handshape preferences for tool naming develop, we turn to the case of homesigners. Homesigners are an important place to look because they, like signers of community signed languages with longer histories, use the manual modality as their primary means of communication. However, homesigners have little to no exposure to an existing sign language and communicate almost exclusively with hearing gesturers in their daily lives.

### 1.2. How Do Homesigners Compare to Communication Partners and Signers?

Homesigners are deaf individuals who have not acquired a signed or spoken language and who innovate gesture systems to communicate with hearing friends and family members. The homesigners in the current study have not had regular contact with each other or with signers of Lengua de Señas Nicaragüense (NSL); each individual has created their own unique system to use with hearing friends and family members (referred to here as "communication partners") (Coppola and Newport 2005; Coppola 2002). Homesign systems more closely resemble sign languages than gestures produced by non-signers (Brentari et al. 2012; Horton et al. 2015). However, since homesigners do not form a linguistic or social group, there is not a large overlap for shared handshape forms even on the individual level, and many homesigners do not have a stable handshape form that they routinely use (Goldin-Meadow et al. 2015). Some trends can be found, such as homesigners using handshape type systematically to distinguish agentive and non-agentive events and additionally homesigners showing a slight preference for Handling Handshape for nominals (Goldin-Meadow et al. 2015). Hearing gesturers in general do not use Object and Handling Handshapes systematically like adult and child homesigners do, but both hearing gesturers and homesigners show a lot of between-subject variability (Brentari et al. 2015).

### 1.3. Why Look at Homesign to Understand Sign Language Emergence?

Studying homesign can help elucidate the emergence of certain structures found in sign languages, such as iconic handshape preference. Some sign languages form when a group of deaf individuals (e.g., homesigners) come together. NSL, for example, came to be after a school was founded allowing deaf homesigners to come together and start converging on a signing system (Senghas et al. 2005; Coppola 2020a). As time went on and more individuals started using the same system, it became more conventionalized, that is, members of the community started sharing similar forms and patterns. The emergence of

NSL as an established language in a matter of decades supports the idea that language can be created, given some time and a receptive community of users (Brentari and Coppola 2013). The people that make up the community matter; in most cases like NSL, they must use the system as their primary form of communication in order for it to conventionalize. This is one major difference between sign languages and homesign; sign languages have been used as primary languages for many people over a long period of time, whereas homesign systems are used predominantly by one individual, which their communication partners use only with them. Even though communication partners use the homesigner's system to communicate with them, they do not use the system in the same way the homesigner uses it, so it may not become conventionalized (this is addressed in more detail in the next section; see also Coppola et al. 2013). Individual homesign family groups have the potential to conventionalize, but, if they do, it is much slower than NSL because of how centralized a homesign system is, given that all interactions involve the homesigner (Richie et al. 2014).

*1.4. Is Conventionalization Possible in Homesign Systems?*

Even though communication partners can use the homesigner's system, there is evidence that they do not always use the same patterns or the same degree of complexity, raising the question of whether homesign systems can become conventionalized. While Nicaraguan hearing gesturers produce gestures similar to some NSL signs, there is evidence of changes in form and meaning, likely mediated by homesigners; however, even over the course of 25 years, NSL still stabilized a lexicon much faster than homesign systems (Coppola 2020b). Homesigning children in Taiwan and the United States typically use similar gesture order, an ergative syntactic pattern in which patients and intransitive actors come before action gestures, while their parents do not follow their children's order and will sometimes put a transitive actor before action gestures, but this is not done consistently (Zheng and Goldin-Meadow 2002). In another group of American child homesigners, the mothers' gestures did not show the same structural regularities compared to their children's gestures; differences in each child's system is related more to the gestural input that the children provide for themselves and less due to any input their mothers may provide (Goldin-Meadow et al. 1984). It seems that communication partners are not enough; in order for conventionalization to happen more rapidly, homesigners must interact with other deaf people using a signing system. For example, in Nebaj, Guatemala, individual homesigners (i.e., those with no interaction with another deaf person) showed weak evidence for the use of patterned iconicity or a preferred handshape type when labeling items, while homesigners who used a shared system with either other deaf family members or deaf peers showed strong evidence for the use of patterned iconicity (Horton 2020).

Not only do communication partners not use the homesigner's system very well, they also do not appear to completely understand it sans context. Homesigners' mothers were significantly worse at comprehending homesign descriptions of vignettes from their deaf adult children than Spanish descriptions from their hearing adult children (Carrigan and Coppola 2017). This study also found that the younger a family member was when they first interacted with their deaf relative, the better their comprehension was; however, Deaf native ASL signers, who were not familiar with the homesign systems but did have lifelong experience perceiving and communicating in the visual modality, were actually the best at comprehending the homesign descriptions. This supports the idea that homesign systems are not completely transparent and that structure within a homesign system is not developed so that a homesigner can be understood by their communication partners, but instead perhaps represents how the homesigner mentally organizes concepts. This result is consistent with the idea that homesign systems are sufficiently similar to languages with longer histories and more developed structure that they also show hallmarks of a sensitive period for acquiring them among those who are exposed to them at different ages (Mayberry and Kluender 2018; Newport et al. 2001).

We focus on iconic handshape preference in the current study and ask whether or not iconic handshape preference is part of an individual system, and whether it is a structure that homesigners and communication partners share with one another. Further, we ask if the degree to which the iconic preference is shared in homesigners' families is stronger than that which occurs in families without a homesigner (in this case, from hearing non-signing Spanish-speaking Nicaraguan families).

*1.5. The Current Study*

By looking at the case of homesigners and communication partners as well as hearing non-signing adults, the current study aims to investigate the development of iconic handshape preferences for tools. Hwang et al. (2017) point out that comparisons across groups can provide "an opportunity to examine possible pathways for grammaticization and conventionalization from emergent to established sign language lexicons and grammars" (p. 578). We investigated several possible sources of handshape preference and conventionalization (Table 1). While signers may show a preference for either Handling or Object handshapes depending on the sign language they use, hearing gesturers (e.g., communication partners and hearing non-signers) typically tend to use Handling handshapes more often (Padden et al. 2015). Although communication partners have demonstrated conventionalization of some types of forms and structures, they often do not utilize it to the same extent as homesigners (e.g., communication partners use unpunctuated repetition in isolation but not in sentences like homesigners do, Coppola et al. 2013). Therefore, the user's relationship with the homesign system may be an important factor. By contrasting homesigning families with a hearing non-signing family, we can see if using a homesign system influences handshape preferences and conventionalization. Similarly, by comparing families with an unrelated group of hearing people, we can address whether communicative familiarity is a factor in handshape preference and conventionalization. We included both chronological age and years of experience with a homesign system as factors that might influence handshape preference and conventionalization, given the findings from Carrigan and Coppola (2017), which indicated that the younger a family member was when they first started interacting with their deaf homesigning relative, the better they understood them. This is also related to the sensitive period for language acquisition which research demonstrates is a relationship between the age of exposure to a language and the proficiency in that language (e.g., Newport 1990; Mayberry and Fischer 1989; Emmorey and Corina 1990). Lexical frequency (i.e., how often a word or sign is used) and type of noun (i.e., whether or not it is an instrument and what type of instrument it is) are additional factors related to the item itself that may also influence preferences and conventionalization.

In Study 1, we analyze participant characteristics, specifically looking at how factors related to the participants (e.g., age, experience with a homesign system) may influence handshape preferences for iconicity (Handling/Object) as well as general conventionality (e.g., average of family's shared handshape preferences regardless of actual handshape type) within and across groups of families with and without homesigners. In Study 2, we analyze item characteristics, specifically investigating how factors related to the stimulus items (e.g., lexical frequency, type of instrument) influence handshape preference and conventionality.

The questions we aim to address in the two studies are as follows:

(1a)  Do homesigners and communication partners tend to express iconicity by using a Handling handshape or an Object handshape?
(1b)  Do members of families with homesigners share this preference with each other?
(1c)  Does a participant's age at the time of test, the age at which they begin using the homesign system, or the number of years they have been using the system affect handshape type and its conventionalization?
(2)   Do some stimulus items elicit higher conventionality in iconic handshape preferences? Which factors are or are not associated with greater conventionality?

**Table 1.** Summary of potential sources of handshape preference and conventionalization.

| Type of Specific Bias | Possible Sources of Handshape Preference | What We Will Look at |
|---|---|---|
| Participant (Study 1) | Using a Homesign System with Others | homesigning families vs. hearing non-signing family |
| | Communicative Familiarity with Others | families vs. unrelated group |
| | Relationship to Homesign System | primary user vs. communication partner vs. none |
| | Age | child/adolescent vs. adult; age at which first started using system; chronological age |
| Item (Study 2) | Lexical Frequency | English, Spanish and ASL word frequencies as proxies for homesign |
| | Type of Instrument | e.g., traditional tool vs. makeup vs. non-tool |

## 2. Materials and Methods

### 2.1. Participants

Participants were recruited between 1996 and 2004 through personal visits to families living in both rural and urban areas of Nicaragua who were recruited via community contacts (see Table 2 for a summary of demographic information; see Gagne (2017) for more detailed information about the homesigners and their linguistic and educational experiences). The participants included eleven deaf homesigners (4 female, 7 male), aged 9 to 35 years at the time of testing. The homesigners had little to no formal education in written or spoken Spanish or Lengua de Señas Nicaragüense (NSL). Homesigners were further classified as either adult homesigners (4 participants, age 26 to 35) or child/adolescent homesigners (7 participants, age 9 to 14).

**Table 2.** Demographic information.

| | N | Age (Years) (Mean, SD) | Gender (% Women) |
|---|---|---|---|
| Homesigners | 11 | 19;1 (9.97) | 36% (4) |
| Communication Partners | 24 | 31;6 (17.1) | 50% (12) |
| Hearing Non-Signers related (4); unrelated (4) | 8 | 31;0 (12.5) | 50% (4) |
| All Participants | 43 | 28;3 (15.4) | 47% (20) |

A second set of participants were the 24 communication partners of these deaf homesigners (12 female, 12 male), who were aged 9 to 64 years at the time of testing. Communication partners (CPs) were defined as hearing family members and friends who had regular contact with and communicated frequently with one of the homesigner participants. All of the communication partners were native Spanish speakers who had no experience with any sign language and who were familiar with the homesign system used in their family.

The third set of participants included 8 hearing non-signing adults (4 female, 4 male) aged 20 to 52 at the time of testing. All of the hearing non-signing adults were native Spanish speakers with no regular experience with any sign language or homesign system.

The participants were grouped into 9 homesigning families (each family had only one homesigner), 1 hearing non-signing family made up of 4 members, and 1 group of 4 unrelated hearing non-signers. Note that the hearing non-signers who were from the same family did not have a homesigner in their family; further, none of the unrelated hearing non-signers had a homesigner in their families. Two homesigners who were in the original

study did not have any communication partners to complete this task and therefore were excluded from group analyses.

## 2.2. Materials and Procedure

Participants were shown a slideshow consisting of photographs of items and asked to label them. Specifically, participants were shown a slide featuring three exemplars of the same tool, such as a hammer, and then were asked to sign what it was. After the participant had finished responding to the current image, the experimenter would move on to the next slide. All data were collected between 2011 and 2012, and each session was videotaped. All participants were tested individually and signed their responses to the experimenter in order to avoid influencing the responses of other members of their family.

The stimuli presented were 27 images of different tools and instruments: 6 items of clothing (pair of shoes, jacket, sock, hat, glove, pants), 8 grooming/cosmetic items (hairbrush, nail file, mascara, comb, hairdryer, nail polish, toothbrush, lipstick), 3 utensils (fork, spoon, knife), 9 handheld tools (scissors, broom, hammer, paintbrush, rake, screwdriver, vacuum cleaner, handsaw, mop), and 1 other handheld item (cellphone). This set of stimuli was also used in Padden et al. (2015). Every participant was shown the stimuli in the same order via the slideshow.

## 2.3. Transcription and Response Types

The participants' signs and gestures were transcribed using ELAN (Wittenburg et al. 2006), a program that facilitates the coding of simultaneous aspects of gesture and sign language production that is aligned with the video content. Each sign was glossed and coded for handshape representation type for both hands. The relevant handshape types were *Handling* (Figure 1a), in which the handshape reflects how one would hold the tool, and *Object* (Figure 1b), in which the handshape depicts the shape or form of the actual tool itself. Other handshape representation types included *Handling–Object-Simultaneous*, in which the participant simultaneously produces a Handling handshape with one hand and an Object handshape with the other hand, and *Handling–Object-Sequential*, in which the handshape sequentially transitions from one representation type to another, which we collapsed into *Handling+Object* (Figure 1c). Signs marked as *Other* (i.e., not specifically in reference to the tool or not iconic) were not included in the analysis.

For each item response, only one handshape type was annotated. If participants made multiple iconic signs/gestures while labeling the item, the response selected for coding was simplified to note just Handling or just Object if all of the signs fell under one type, or Handling+Object if both types of signs were used. Therefore, each participant had a maximum response of 27 handshapes, which we were then able to use to calculate the percentage of iconic handshape types in order to determine preferences.

Finally, we describe the types of responses participants produced. Most people produced single gestures/signs, with the exception of adult homesigners who produced multiple signs more often (Table 3). Although participants were shown all 27 items and asked to label them, a few items were not familiar to the participants (e.g., vacuum cleaner) and they did not produce a response to them. Factoring in all of the participants, the total expected number of responses was 1161. However, we only included a total of 1035 responses, because not every participant produced a response for every item (e.g., some participants did not recognize the vacuum cleaner) or did not produce a relevant iconic response (e.g., the response was a pointing gesture).

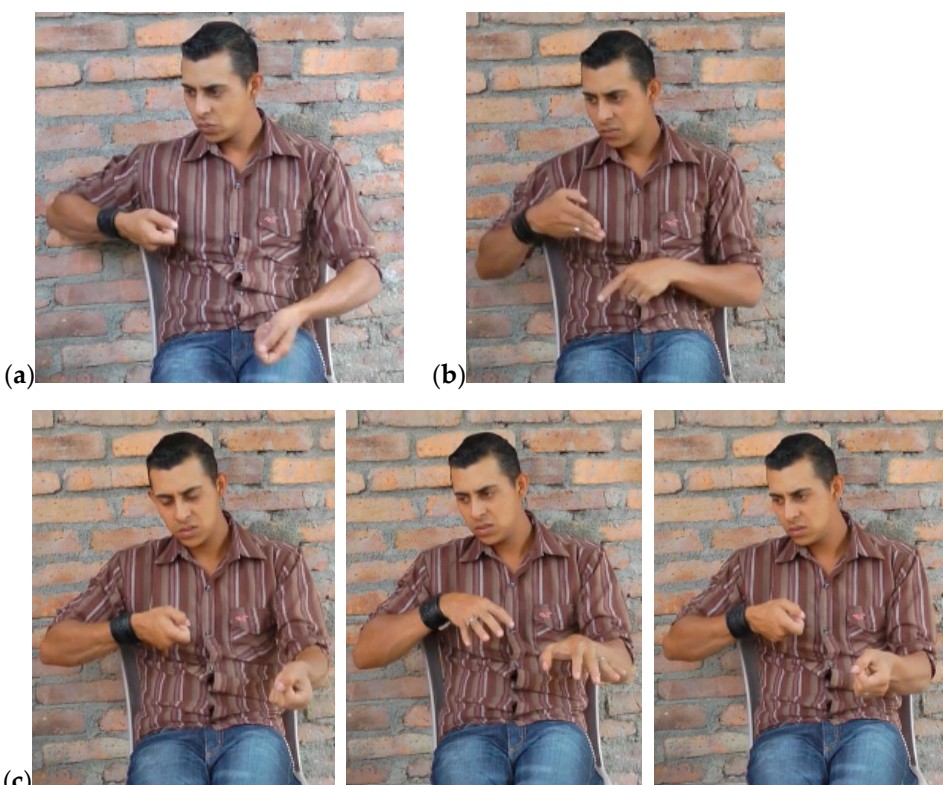

**Figure 1.** Examples for responses of iconic handshape types: (**a**) Handling [mop], (**b**) Object [hand saw], and (**c**) Handling+Object [rake]. See supplementary materials to view video clips of responses.

**Table 3.** Percentage of responses that were a single sign/gesture versus multiple signs/gestures for each group. Most common response type for each group is bolded.

|  | Responses Consisting of Single Gesture/Sign (Mean, SD) | Responses Containing Multiple Gestures/Signs (Mean, SD) |
|---|---|---|
| Adult Homesigners | 32% (0.34) | **68%** (0.34) |
| Child/Adolescent Homesigners | **92%** (0.05) | 8% (0.05) |
| Communication Partners | **80%** (0.26) | 20% (0.26) |
| Hearing Non-Signers | **93%** (0.14) | 7% (0.14) |

### 3. Results

*3.1. Study 1: Participant Characteristics*

3.1.1. Do Homesigners and Communication Partners Tend to Express Iconicity by Using a Handling Handshape or an Object Handshape?

First, we want to clarify that we did not expect to see a strong overall preference for one iconic handshape type over the other, because many other features outside of the participant can influence preferences, primarily item characteristics, which we address in Study 2. While we do report overall preferences (i.e., the proportion of Handling, Object and combined Handling+Object responses across all items for each participant), it is important to keep in mind that features of specific items may also influence handshape preferences and are not captured by looking at overall preference.

There appears to be no universal preference across types of participants (e.g., homesigners or their communication partners) or across participant groups for one of the handshape types. Indeed, individuals varied greatly in whether they showed a preference, as well as which handshape type they preferred when they demonstrated a preference. We proceeded

to undertake more detailed analyses of potential patterns within participant groups. Of the 11 homesigners, 5 showed a Handling preference, 4 of which were child/adolescent homesigners, as determined by binomial distribution tests (Table 4). In order to carry out the binomial distribution tests, we only compared two categories: Handling vs. Object; on trials in which a participant used a combined Handling+Object form, the response was counted as half a case of Handling and half a case of Object. For example, for a participant who produced 12 Handling handshapes, 7 Object handshapes, and 8 combined Handling+Object forms, their responses were simplified as 16 cases of Handling and 11 cases of Object handshape in the binomial test. Of the 24 communication partners, 5 showed a reliable Handling preference, and 2 showed a reliable Object preference. Of the 8 hearing non-signing adults, 1 showed a reliable Handling preference. Overall, the majority of participants did not show a handshape preference. Of the participants who did show a preference, a majority had a preference for Handling handshapes.

**Table 4.** Mean iconic handshape preferences of child/adolescent and adult homesigners, communication partners (CPs) and hearing non-signers (related and unrelated). H+O stands for combined Handling+Object handshape. See Appendix A for individual and group preferences.

| Participant Type | Handling | Object | H+O |
|---|---|---|---|
| Child Homesigners | 67% | 29% | 4% |
| CPs of Child Homesigners | 53% | 43% | 4% |
| Adult Homesigners | 47% | 26% | 27% |
| CPs of Adult Homesigners | 56% | 34% | 11% |
| Hearing Family | 51% | 42% | 8% |
| Hearing Unrelated | 54% | 37% | 9% |

3.1.2. Do Members of Families with Homesigners Share This Preference with Each Other?

Next, we looked at group preferences to assess how communicative familiarity and using a homesign system with others might influence handshape preferences. With regard to the homesigning families, 3 families showed a Handling preference, and the other 6 families showed no clear preference (see Appendix A). Additionally, neither of the groups of hearing people who had no regular communication with a homesigning family member (i.e., the all-hearing family and the group of unrelated hearing people) showed a clear preference. Note that, of the homesigning families that showed an overall Handling preference, two were families with child or adolescent homesigners, and one was a family with an adult homesigner.

We also examined the conventionality of iconic handshape preferences; instead of looking specifically at handshape type, we calculated the likelihood of participants within a group producing the same handshape type, at the item level, regardless of whether the handshape was Handling or Object. In order to calculate the conventionalization for each group, we compared each family member's responses to each other in a pairwise fashion. For each item, when two family members produced the same handshape type (e.g., both used Handling), the pair was assigned 1 point. If they produced different responses (e.g., one used Handling and one used Object), they were assigned 0 points. If one family member used a combination form (e.g., one used Handling+Object, while one just used Handling), they were assigned 0.5 points. For each family member pair, point values were totaled, and the percentage of similar response types was calculated. Once the similarity percentages were calculated for each pair of participants in each family group, the percentages were averaged for the entire group. For example, in Adult Homesigner 4's family, the homesigner and brother produced the same handshape type on 60% of items, the homesigner and mother produced the same handshape type for 63% of items, and the brother and mother produced the same handshape type on 56% of items, leading to an average of 60% conventionalization for the family. In other words, the members of Adult

Homesigner 4's family shared handshape preferences with one another, on average, 60% of the time. Overall, the conventionalization of iconic handshape type (i.e., Handling or Object) ranged from 48% to 71% across the groups (Table 5). Members of adult homesign families ($n_1$ = 4) were significantly more conventionalized than child/adolescent homesign families ($n_2$ = 5) (U = 1, $p < 0.05$, Mann–Whitney U test for small Ns).

**Table 5.** Average conventionalization of all family members for each group.

| Group | Conventionalization (Mean, SD) |
|---|---|
| Child Homesigner 1 | 61% (0.12) |
| Child Homesigner 2 | 71% (0.10) |
| Child Homesigner 3 | 54% (0.14) |
| Child Homesigner 5 | 48% (0.15) |
| Child Homesigner 6 | 65% (0) |
| Adult Homesigner 1 | 70% (0.10) |
| Adult Homesigner 2 | 66% (0.03) |
| Adult Homesigner 3 | 69% (0.11) |
| Adult Homesigner 4 | 60% (0.40) |
| Hearing Related | 59% (0.08) |
| Hearing Unrelated | 70% (0.17) |

3.1.3. Does a Participant's Age at the Time of Test, Age at Which They Begin Using a Homesign System, or the Number of Years They Have Used the System Affect Handshape Type and Conventionalization?

We assessed the relationships between the communication partners' conventionalization and (i) age at time of testing, (ii) years of experience with a homesign system, and (iii) age of first exposure to a homesign system. We used age two years as the "starting point" for homesign systems and based our calculations of age of exposure to the homesign system and years of experience with a homesign system on that value (see Carrigan and Coppola 2017). For the homesigners, years of experience using a homesign system corresponded to their age minus 2 years; for family members, years of experience using a homesign system corresponded to the family member's age, minus the homesigner's age, minus 2 years (the initial years of homesign development). Similarly, the variable of age of exposure to the homesign system also took this two-year period of initial, early homesign development into account. For example, the mother of a homesigner (aged 26) who was 54 at the time of testing would have been exposed to the homesign system at age 30 (i.e., (54 − 26) + 2 = 30). Younger siblings were assigned a value for age of exposure corresponding to their age (assuming they were born two or more years after the homesigner). Note that the age of the homesigner at the time of testing is a good approximation of how long the family has been co-constructing and using the homesign system.

We found moderate inverse correlations between conventionalization and age at the time of testing ($r_s$ = −0.49, $p < 0.05$, Spearman's rho) and between conventionalization and years of experience using a homesign system ($r_s$ = −0.42, $p < 0.05$). That is, higher conventionalization was associated with being younger when tested and with fewer years of experience using a homesign system. A weak inverse correlation was found between conventionalization and age of first exposure ($r_s$ = −0.22, $p > 0.05$), showing that higher conventionalization was somewhat associated with being exposed to a homesign system from a young age. While these findings do not necessarily contradict one another, they do raise some questions which are considered in the discussion section. Linear regressions revealed that the communication partners' age at testing (t = −2.51, $p < 0.05$; F(1, 22) = 6.35, $p < 0.05$, $R^2$ = 0.22) and their years of experience interacting with a homesigner (t = −2.33,

$p < 0.05$; $F(1, 22) = 5.41$, $p < 0.05$, $R^2 = 0.20$) each significantly predicted conventional-ization. The communication partners' age of first exposure to homesign did not predict conventionalization.

We also found differences across participants in how they used both Handling+Object handshapes within a single response. Adult homesigners produced this Handling+Object response type more often (M = 27% of items, SD = 0.18) compared to child homesigners (M = 6%, SD = 0.02), communication partners (M = 7%, SD = 0.09), and hearing non-signers (M = 8%, SD = 0.08).

### 3.2. Study 2: Item Characteristics

We now turn to the item analysis, to determine which individual objects and which semantic classes of object (clothing, grooming/cosmetic items, utensils, and handheld tools) exhibited a greater degree of conventionalization.

#### 3.2.1. Do Some Stimulus Items Elicit Higher Conventionality in Iconic Handshape Preferences?

In order to examine item-specific biases, we conducted binomial distribution tests and found that many of the responses elicited by certain items were not at chance (50%) and, in fact, were biased towards one or the other iconic handshape class. Of the items not at chance, more items were more likely to have a Handling preference than Object (Table 6), and tools were more likely to exhibit a higher degree of conventionalization. None of the items at chance were classified as traditional tools.

**Table 6.** Binomial tests revealed that overall, 12 items were more likely to elicit a Handling handshape and 4 items were more likely to elicit Object handshapes, while 11 items were statistically at chance.

| Handling Bias (12) | Object Bias (4) | Dependent on Chance (11) |
|---|---|---|
| Pants | Scissors | Nail File |
| Hammer | Knife | Cell Phone |
| Spoon | Handsaw | Hat |
| Sock | Paintbrush | Jacket |
| Mop | | Mascara |
| Broom | | Nail Polish |
| Hairbrush | | Toothbrush |
| Screwdriver | | Lipstick |
| Vacuum Cleaner | | Rake |
| Hair Dryer | | Glove |
| Fork | | Shoe |
| Comb | | |

#### 3.2.2. Which Factors Are or Are Not Associated with Greater Conventionality?

The participant families' degree of conventionalization varies; however, certain items may lend themselves to being consistently gestured or signed with one type of iconic handshape over the other. Participants converged on a handshape type for 22 of 27 items; that is, over 50% of the group members used the same handshape type. Traditional tools all had higher conventionalization (all >70%), while makeup items all showed lower conventionalization (Figure 2). As expected, the items that have higher conventionalization are also the items that are more likely to have a handshape bias.

We also considered lexical frequency as a possible motivation for conventionalization. Because we did not have homesign frequency counts, we used ASL, English, and Spanish word frequencies as proxies. We obtained measures of ASL sign frequency, English word frequency, and Spanish word frequency from Corpus del Español (Davies 2015). While ASL, English, and Spanish word frequencies were all moderately to strongly correlated with each other, none of the word frequency measures were correlated with the degree of conventionalization observed for a particular item. While these functional/semantic group-

ings did reveal some trends toward greater (e.g., traditional tools) or lesser (e.g., makeup items) conventionalization, they were fairly weak and do not seem to be explanatory.

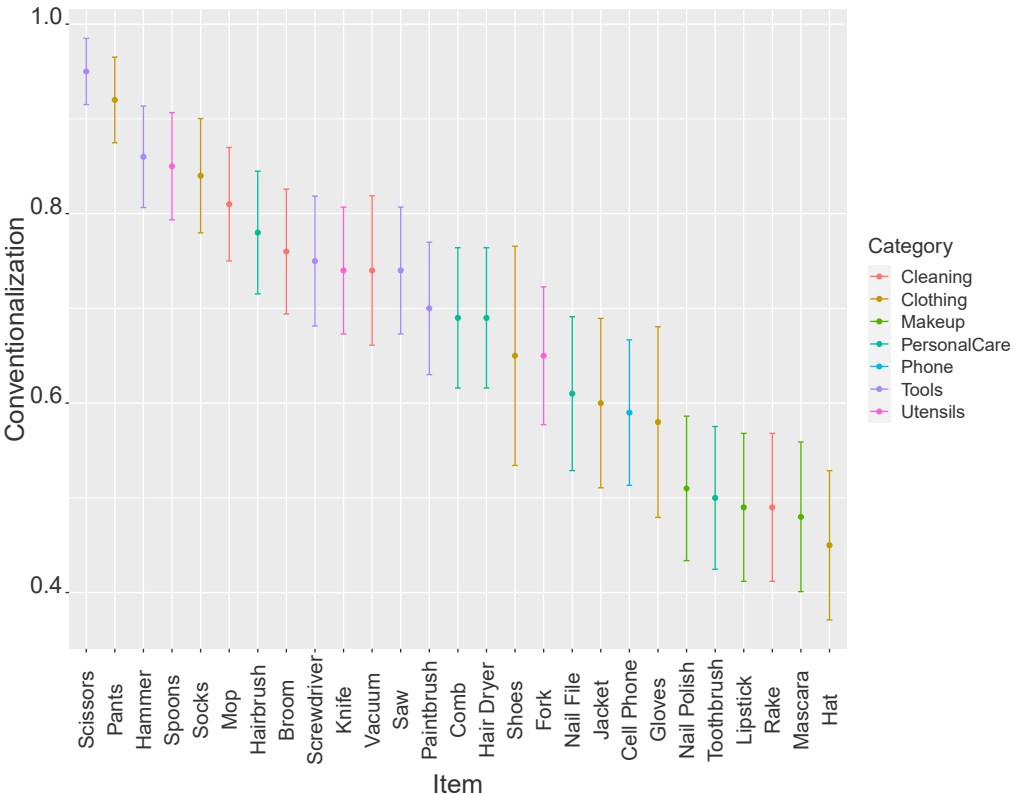

**Figure 2.** Items ordered by overall conventionality of handshape type across groups. Each category of object is shown in a different color. Dots represent the percentage of participants who used the most common handshape type for each item (Handling or Object). Lines represent standard error.

## 4. Discussion

In this paper, we used the phenomenon of iconic handshape contrasts (Handling vs. Object), a distinction used grammatically by many sign languages around the world, as a lens through which to examine the development of conventionalization in emerging sign languages. Specifically, we examined iconic handshape preferences in deaf homesigners and their hearing communication partners when naming objects used as tools (see Table 7 for a summary). Study 1 revealed that participant characteristics influence preference for handshape type. Among families with homesigners, we detected no universal handshape preference in either the homesigners or the communication partners, though several participants (many of them gesturers) tended to use more Handling forms. Within families with homesigners, we also observed variable levels of shared preferences for handshape type and the degree of conventionalization for handshape type. Among homesigners' families, time using the homesign system seems to be important, because adult homesigners' families demonstrated a higher degree of conventionalization than child homesigners' families. Study 2 found that traditional tools (e.g., scissors, hammer) tended towards higher conventionality than other items that were not traditional tools (e.g., mascara, hat). Proxy measures of lexical frequency did not show any correlation with the degree of conventionalization.

**Table 7.** Results summary.

| Type of Specific Bias | Possible Sources of Handshape Preference | What We Found |
|---|---|---|
| Participant (Study 1) | Using a Homesign System with Others | Homesigning families varied just as much as the hearing family |
| | Communicative Familiarity with Others | Families varied just as much as the unrelated group of hearing people |
| | Relationship to Homesign System | No differences found between homesigners, CPs, and non-signers |
| | Age | Chronological age and years of experience predicted conventionalization; adult homesigners produced more Handling+Object responses |
| Item (Study 2) | Lexical Frequency | Proxy measures not related to conventionalization |
| | Type of Instrument | Traditional tools often more highly conventionalized than other items (e.g., makeup) |

These findings suggest that there is no universal handshape preference among the different participant groups when naming tools, and that similarity of handshape type varies even within families. Both individuals and groups varied in terms of iconic handshape preference. The fact that a family did or did not have a homesigner or, taken as an isolated factor, communicative familiarity (i.e., families versus unrelated group of people), does not promote similarity or conventionalization of iconic handshape type; however, there is evidence that, among families with a homesigner, the longer the family uses the system, the more conventionalized it becomes (adult homesign families are more conventionalized than child homesign families). It may be that individual preference initially drives this type of iconic handshape preference.

*4.1. Variation in Iconic Handshape Preference and Conventionalization*

While more individuals (i.e., homesigners, communication partners, and hearing non-signing adults) produced more Handling than Object handshapes in their responses, as a group, fewer than half showed an overall Handling preference. In other words, while 25 of the 43 participants (58%) showed an individual Handling preference, only 4 of the 11 groups (36%) showed a clear overall Handling preference. The fact that a majority of the data points were from communication partners (i.e., gesturers) and that responses were overall more likely to use Handling handshapes aligns with the finding that generally gesturers, compared to signers, prefer Handling handshapes when labeling tools (Padden et al. 2015). The next step will be to see whether homesigners (particularly adult homesigners) and communication partners use the same specific handshapes, instead of merely the same type of handshape iconicity. In line with existing research, homesigners' responses exhibit more finger and joint complexity than the responses of hearing non-signers (Brentari et al. 2012; Coppola and Brentari 2014), but explicit comparisons of homesigners with communication partners have yet to be completed.

The fact that conventionalization varied greatly among homesigning families, ranging from 48% to 71% conventionalization, suggests that this phenomenon of patterned iconicity may emerge later. This is supported by evidence that families with adult homesigners were overall more highly conventionalized than families with child/adolescent homesigners. In groups where this was not the case, such as the family of child homesigner 2 and the group of unrelated hearing people (both of whom had some of highest conventionalization rates), something else might be influencing handshape conventionalization. We included the hearing non-signing family and the group of unrelated hearing non-signers in order to pull apart

the influence of having a homesigner in a family versus generally interacting within a family; however, it was somewhat surprising that the unrelated hearing non-signers displayed such high conventionalization. Note that our measure of conventionalization does not take into account the complexity of the handshapes produced. Thus, the hearing people in our study may be producing the most straightforward and least complex handshape forms. In their context as hearing people who do not regularly communicate with a deaf person, they are not burdened with additional iconicity demands (e.g., using patterned iconicity systematically) and may therefore consider fewer, less complex handshape options. Since hearing non-signing gesturers already tend to both produce less finger and joint complexity than homesigners (Brentari et al. 2012; Coppola and Brentari 2014) and prefer Handling handshapes (Padden et al. 2015), this may partly be responsible for the convergence. In other words, the Handling handshape preference may also be generated independently in each individual rather than due to convergence in a group. In addition, it follows that they would exhibit restricted options for types of handshapes, and their productions, therefore, may appear more conventionalized, even if little actual conventionalization has occurred with the apparent similarity merely reflecting similar strategies. In other words, although the unrelated hearing non-signers may appear to be highly conventionalized, they are likely using the simplest forms at their disposal and, given their limited handshape options, just happen to be using the same simple handshape types.

This line of reasoning may also explain the finding that the less experience a communication partner has with a homesign system, the more conventionalized they are; essentially, these communication partners may be using the most straightforward or simplest approach without much true conventionalization happening (e.g., everyone converging on a shared complex handshape versus each person producing the simplest responses, which coincidentally happen to be similar). Previous work by Singleton et al. (1993) comparing the productions of the homesigner called David with those of his hearing sister showed that her gestures more closely resembled those of non-signers rather than those of his homesigning brother. Frequent and prolonged interaction between communication partners and homesigners does not appear to be enough to conventionalize gestures in a homesign system. As David was documented correcting his sister's gesture forms, so has Coppola observed adult homesigners correcting their family members, providing similar evidence of standards of form in homesign systems that communication partners do not always pick up on. The relationship between age and experience (discussed more below) could perhaps be explained by the association of younger age and simpler forms, and less to do with practice using the system. Uniformity due to simple forms may be masquerading as conventionalization.

### 4.2. Age and Experience as Factors in Conventionalization

We noticed some trends related to the age of the homesigner, mainly that, overall, adult homesigners used the combined Handling+Object form more than child homesigners, and that families with adult homesigners were significantly more conventionalized than families with child/adolescent homesigners. This tendency for a combined Handling+Object form to be produced by adult but not child homesigners is related to the finding that adult homesigners more often produced multiple signs or gesturers for single responses compared to child/adolescent homesigners, who typically produced just one sign or gesture per response. While the Handling+Object form clearly made up a portion of the adult homesigners' multiple signs/gestures per response, adult homesigners also tended to produce multiple of the same type of handshape in a single response, which child homesigners did not do frequently. Adult homesigners' more common use of a combined Handling+Object form has also been observed in independent groups of adult Nicaraguan signers and child homesigners in Nicaragua. Martin et al. (Forthcoming) found that 10 out of the 11 signing Nicaraguan adults in their study produced a combined Handling+Object form at a similar level to that of the adult homesigners in our study, but it was not robust in the responses produced by child homesigners or hearing gesturers.

This suggests that this combined form, which is present in adult homesigners and persists among Nicaraguan signers from the second and third cohorts (that is, among signers who entered the community after 1983), albeit at low levels, requires maturational time to develop and is not a response to a communicative context experienced by homesigners in which they are concerned about not being understood.

We also would like to note that we specifically refer to this response type as a combined form, not a compound form, because we did not rigorously assess each response to determine if they were true compound signs. Published criteria for identifying compound signs are limited, and the classification is commonly determined by intuition or judgments (i.e., Meir et al. 2010) or by comparing two-sign combinations in which the first stem is reduced to stand-alone versions of the same signs (Liddell and Johnson 1986). Note that existing criteria are difficult to implement in emerging languages, especially homesign systems, in which the forms themselves may be in flux. Further, traditional acceptability judgments and intuitions are difficult (though not impossible) to elicit from homesigners. In addition, the data collected in this sample did not allow the opportunity to compare single signs and two-sign combinations. Since this combined category included both simultaneous and sequential Handling+Object forms, we decided to refer to them as combined forms rather than teasing out which could be compounds and which were not. In some instances, participants produced a "sandwich form" such as Handling, Object, then Handling again. For example, Adult Homesigner 1 produced a Handling+Object + Handling form when labeling the rake. This strategy has the advantage of making it clear that this is the name of the item, not describing the action carried out by the item. These combined forms tend to be produced very quickly, with little to no pause between the signs. While this category of signs may be a candidate for compound forms, that analysis is outside of the scope of this paper.

Another age-related trend was that, among the families that showed a general preference for producing Handling handshapes, most were families with child homesigners. We did observe a preference for Handling handshapes among the communication partners of one of the adult homesigners; this family had the greatest number of communication partners (5, as opposed to 2 to 3 communication partners like the other adult homesigners). As hearing gesturers, who tend to produce more Handling responses, these communication partners drive up the mean of the overall use of Handling handshapes within the family. Might regular interactions with communication partners influence homesigners to be less consistent with their handshape preferences? Goldin-Meadow et al. (2015) found that homesigners' first inclination, before elaborating more extensively for their hearing communication partner, was to produce a pattern comparable to the pattern produced by ASL and NSL signers (i.e., more consistent). However, communicating with hearing partners, who sometimes struggle to comprehend homesigners' productions (as demonstrated in Carrigan and Coppola 2017), may make homesigners' systems appear less structured. This finding might provide an alternate explanation to the phenomenon of adult homesigners using the combined Handling+Object form more often than child homesigners; perhaps they are used to having to elaborate on initial signs when communicating with hearing partners, therefore they may be inclined to use a combined Handling+Object form in order to be more clear and to avoid having to repeat or elaborate. However, this explanation does not account for the persistence of this combined form among NSL signers over 30 years after the emergence of the community. Clearly, more research is warranted.

Homesign systems have the potential to conventionalize, albeit more slowly than a shared sign language used by a deaf community, since each system is only used primarily by a single homesigner (Richie et al. 2014). We found that participants' age at time of testing and their years of experience using a homesign system were negatively correlated with conventionalization. The younger a communication partner was and the fewer years they spent using a homesign system, the higher the degree of conventionalization. Given Goldin-Meadow et al. (2015) findings that homesigners' subsequent responses are much more variable than their first responses, it is possible that interacting with

communication partners, who are not using the homesign system as their primary system, may hinder or even deconventionalize certain homesign patterns. It is also possible that, among a relatively small number of family members who interact regularly and whose communication is embedded with specific contexts, the lack of a conventionalized form is less of an issue. Future research should investigate how age and amount of time a person has been using a system influence conventionalization in homesign systems, in addition to other factors such as communicative closeness and the amount of time that communication partners actually spend interacting with a homesigner.

### 4.3. Item-Specific Biases and Proxy Measures

Although conventionalization varied within and across family groups, we did find that certain types of items were, overall, more highly conventionalized than others. Specifically, tools (e.g., scissors, hammer) had higher conventionalization than makeup (e.g., mascara, lipstick) or items of clothing. Similar findings of tools having higher convergence than other categories of items such as animals or food for homesigners from Nebaj, Guatemala, have also been reported (Horton 2018). This observation led us to consider word or gesture frequency; since not everyone uses or converses about makeup, perhaps that is why makeup items had generally lower conventionalization than more widely used items (e.g., scissors, spoons, socks). Unfortunately, it was not feasible to obtain a homesign frequency measure for each item for each family, so we used word frequency measures from ASL, English, and Spanish (all of which were correlated with each other) as a proxy for the frequency of use of such items in homesign systems. We found that conventionalization was not correlated with any measure of word frequency (or ASL iconicity); however, this does not necessarily mean word usage is not at all related to homesign conventionalization. Instead, it is possible that none of the proxy measures used here are actually representative of the frequency of such items in homesign systems. In order to truly understand the relationship between conventionalization and frequency, future studies should obtain iconicity ratings and a measure of homesign frequency from those using the homesign system and directly compare the conventionalization and usage in the system.

### 4.4. More Questions and Future Directions

In future work, we will study whether homesigners and communication partners use the same specific handshapes, in addition to the same type of handshape iconicity (i.e., Handling or Object). This would suggest that there is something shared among homesigners and communication partners that is at the level of phonetic form. Previous research (Brentari et al. 2012) showed that homesigners' responses show higher selected finger complexity than gesturers' responses; however, communication partners' productions have not been analyzed. We do not yet know if the productions of communication partners align more with the complexity levels of homesigners or with those of gesturers. Very preliminary analyses of the current data suggest that families that share a general iconic handshape type do not always produce the same specific handshape, for example, in response to the stimulus item eliciting 'saw', family members may all produce different versions of an Object handshape (e.g., B-handshape, H-handshape, or 1-handshape which all resemble a saw). To illustrate, Adult Homesigner 1 and his brother used the same general handshape type (i.e., Handling, Object, Handling+Object) for 19 items but only used the exact same specific handshape for 7 of those items (37%). In contrast, two unrelated hearing non-signers also used the same iconic handshape type for 19 items but used the same exact handshape for 12 of those items (63%). Further analysis will investigate whether the reported conventionalization of iconic handshape preferences is related to the conventionalization of specific handshapes.

Another question for future work is: Do conventionalization and similar handshape preferences actually improve comprehension between homesigners and communication partners? Previous research suggests that communication partners are not very good at comprehending homesign utterances that are shown to them without any context. Younger family members (e.g., siblings, especially younger ones) scored better on comprehension than older family members (e.g., parents). Deaf native ASL signers, who were not familiar with the homesign systems but did have lifelong experience perceiving and communicating in the visual modality, were actually the best at comprehending the homesign descriptions (Carrigan and Coppola 2017). Perhaps using similar handshape patterns could facilitate homesign comprehension and should be investigated further. It would also be useful to see how consistent personal handshape preferences are by gathering longitudinal data using these stimuli.

## 5. Conclusions

Investigating Handling–Object handshape contrasts and conventionalization in homesigners, hearing communication partners, and hearing non-signing adults has offered some insight into how this type of patterned iconicity develops. While there was a great deal of variation in handshape type preference, overall, gesturers tended to use more Handling forms. Age was the most influential participant characteristic, as the age of the homesigner and communication partner as well as the communication partners' years of experience using the homesign system were significant factors related to conventionalization. Although proxy measures of lexical frequency did not appear to be related to the degree of conventionalization, the instrument type did seem to make a difference to conventionalization. While families with adult homesigners were more conventionalized compared to families with child/adolescent homesigners, they still did not achieve anything close to full conventionalization. Compared to the trajectories in emerging sign languages, these handshape preferences and conventionalization do not seem to reliably appear within two to three decades of use in a family unit consisting of a single homesigner among hearing family members, when only the one deaf person uses the system as a primary system. Looking at the transition from homesign to Nicaraguan Sign Language will also shed light on the role of community participation in the development of iconic handshape preferences. Future research should aim to investigate how individual biases, item-specific biases, and shared/limited usage of a communication system influence these outcomes, both at the level of handshape class and at the level of specific handshape.

**Supplementary Materials:** The following supporting information are available online at https://www.mdpi.com/article/10.3390/languages7030156/s1, Video S1a: Handling [mop]; Video S1b: Object [handsaw]; Video S1c: Handling+Object [rake]; Video H+O-SIM [mascara].

**Author Contributions:** Conceptualization, D.B. and M.C.; methodology, D.B. and M.C.; formal analysis, M.Q.; investigation, M.C.; resources, M.C and D.B.; writing—original draft preparation, M.Q.; writing—review and editing, M.Q., D.B. and M.C.; visualization, M.Q.; supervision, D.B. and M.C.; funding acquisition, D.B. and M.C. All authors have read and agreed to the published version of the manuscript.

**Funding:** This research was funded by NSF BCS-0547554 and BCS-1227908 (to D.B. and S. Goldin-Meadow) and by NIH P30 DC010751 (to M.C. and D. Lillo-Martin).

**Informed Consent Statement:** Informed consent was obtained from all subjects involved in the study.

**Data Availability Statement:** Datasets from this project are available on OSF at the following link: https://osf.io/db7yk/?view_only=1cf74efc936f45828abfc52fb22c3f9a.

**Acknowledgments:** We thank all of the homesigners, their friends and family members, and the hearing adults in Nicaragua for allowing us to learn from them. We are grateful to the directors and staff of the Center for Special Education in Estelí for their support of this project. Julia Adell, Emily Carrigan, and Deanna Gagne assisted with data collection, and Claudia Molina and Leybi Tinoco graciously provided data collection support in the field in Nicaragua. Tiffani Brophy and Julia Adell coded the video recordings. We also thank Laura Horton for providing feedback on prior drafts.

**Conflicts of Interest:** The authors declare no conflict of interest.

## Appendix A

Iconic handshape preferences of child/adolescent and adult homesigners (HS), communication partners (CPs), and hearing non-signers (related and unrelated). Includes individual and group preferences. Family groups are ordered by the homesigner's age at time of testing; the groups of hearing participants are at the end and are also ordered by participant age. Participant preferences that were significantly above chance have an asterisk. Means for each type of participant are also included. (Note: Adult Homesigner 3 has been referred to in previous research as Adult Homesigner 5; similarly, Adult Homesigner 4 has been referred to previously as Adult Homesigner 3.)

| Group | Age (Years) | Handling Responses | Object Responses | Handling+Object Responses |
|---|---|---|---|---|
| Child HS 1 | 9 | 82% * | 14% | 5% |
| Sister | 14 | 67% | 33% | 0% |
| Mother | 26 | 36% | 60% | 4% |
| Grandmother | 50 | 54% | 46% | 0% |
| *Mean* | | **64%** | **28%** | **8%** |
| Child HS 2 | 12 | 64% | 28% | 8% |
| Father | 31 | 75% * | 13% | 13% |
| Mother | 31 | 52% | 44% | 4% |
| *Mean* | | **64%** | **28%** | **8%** |
| Child HS 3 | 12 | 36% | 59% | 5% |
| Brother | 9 | 6% | 89% * | 6% |
| Friend | 12 | 74% | 26% | 0% |
| Father | 37 | 61% | 35% | 4% |
| *Mean* | | **47%** | **50%** | **3%** |
| Child HS 4 | 12 | 76% * | 24% | 0% |
| Child HS 5 | 13 | 80% * | 13% | 7% |
| Brother | 13 | 55% | 45% | 0% |
| Mother | 33 | 17% | 83% * | 0% |
| Father | 52 | 63% | 31% | 6% |
| *Mean* | | **52%** | **45%** | **3%** |
| Child HS 6 | 13 | 81% * | 15% | 4% |
| Brother | 14 | 75% * | 10% | 15% |
| *Mean* | | **78%** | **13%** | **9%** |
| Child HS 7 | 14 | 53% | 47% | 0% |
| *Overall mean, Child HS* | | **67%** | **29%** | **4%** |
| *Overall mean, CPs of Child HS* | | **53%** | **43%** | **4%** |

| Group | Age (Years) | Handling Responses | Object Responses | Handling+Object Responses |
|---|---|---|---|---|
| Adult HS 1 | 26 | 69% * | 23% | 8% |
| Niece | 9 | 77% * | 23% | 0% |
| Girlfriend | 19 | 62% | 35% | 4% |
| Brother | 28 | 73% * | 23% | 4% |
| Mother | 54 | 58% | 35% | 8% |
| Father | 64 | 50% | 42% | 8% |
| *Mean* | | **64%** | **29%** | **7%** |
| Adult HS 2 | 30 | 36% | 40% | 24% |
| Sister | 23 | 73% * | 19% | 8% |
| Brother | 25 | 38% | 29% | 33% |
| Mother | 46 | 38% | 38% | 23% |
| *Mean* | | **47%** | **32%** | **22%** |
| Adult HS 3 | 34 | 56% | 20% | 24% |
| Brother | 19 | 48% | 44% | 8% |
| Sister | 29 | 59% | 41% | 0% |
| *Mean* | | **54%** | **35%** | **11%** |
| Adult HS 4 | 35 | 26% | 22% | 52% |
| Brother | 44 | 64% | 28% | 8% |
| Mother | 61 | 27% | 50% | 23% |
| *Mean* | | **38%** | **33%** | **28%** |
| *Overall mean, Adult HS* | | **47%** | **26%** | **27%** |
| *Overall mean, CPs of Adult HS* | | **56%** | **34%** | **11%** |

| Group | Age (Years) | Handling Responses | Object Responses | Handling+Object Responses |
|---|---|---|---|---|
| Hearing Related 1 | 23 | 36% | 60% | 4% |
| Hearing Related 2 | 26 | 62% | 35% | 4% |
| Hearing Related 3 | 48 | 31% | 54% | 15% |
| Hearing Related 4 | 52 | 74% * | 19% | 7% |
| *Mean* | | **51%** | **42%** | **8%** |
| Hearing Unrelated 1 | 20 | 56% | 32% | 12% |
| Hearing Unrelated 2 | 22 | 56% | 20% | 24% |
| Hearing Unrelated 3 | 23 | 65% | 35% | 0% |
| Hearing Unrelated 4 | 34 | 38% | 63% | 0% |
| *Mean* | | **54%** | **37%** | **9%** |

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
