# Peer review of "Conventionalization of Iconic Handshape Preferences in Family Homesign Systems"

_languages, doi:10.3390/languages7030156_

Round 1
Reviewer 1 Report
This article contributes to the literature on conventionalization of patterned iconicity, on object vs handling, on homesigning and family networks, and especially on the interest of combined Object+Handling forms (and sandwiched forms!).
The methodology for calculating conventionalization is sound, and in general the quantitative analyses presented here are rigorous and support the overall interpretations.
Of greatest interest to myself was the distinction hinted at on line 538 of a combined form vs a compound form. What is the distinction and how can other researchers make use of or apply it?
The future work addressed on line 608 is extremely intriguing and one could only wish that it some pilot results along these lines could be included within the current paper, to support the results in terms of phonetic form.
I found the first sentence vague "it is unclear where the conventionalization of iconic handshape preferences originates". What does this mean, where it originates? Isn't conventionalization part of all language change?
Overall, this is a solid contribution, with original novel empirical data and clear consequences for the literature, and is essentially ready for publication as is.
Author Response
- Of greatest interest to myself was the distinction hinted at on line 538 of a combined form vs a compound form. What is the distinction and how can other researchers make use of or apply it?
We agree that this question is extremely interesting; in the revised text (pasted below) we elaborated on why we did not assess whether the combined forms could be considered compounds on page 15:
“We also would like to note that we specifically refer to this response type as a combined form, not a compound form, because we did not rigorously assess each response to determine if they were true compound signs. Published criteria for identifying compound signs are limited and is commonly determined by intuition or judgments (i.e., Meir et al., 2010) or by comparing two-sign combinations in which the first stem is reduced to stand-alone versions of the same signs (Liddell & Johnson, 1986). Note that existing criteria are difficult to implement in emerging languages, especially homesign systems, in which the forms themselves may be in flux. Further, traditional acceptability judgments and intuitions are difficult (though not impossible) to elicit from homesigners. In addition, the data collected in this sample did not allow the opportunity to compare single signs and two-sign combinations. Since this combined category included both simultaneous and sequential Handling+Object forms, we decided to refer to them as combined forms rather than teasing out which could be compounds and which were not.”
- The future work addressed on line 608 is extremely intriguing and one could only wish that some pilot results along these lines could be included within the current paper, to support the results in terms of phonetic form.
We added a brief discussion of very preliminary analyses regarding shared specific handshape type on page 16:
“Very preliminary analyses of the current data suggest that families who share a general iconic handshape type do not always produce the same specific handshape; for example, in response to the stimulus item eliciting ‘saw’, family members may all produce different versions of an Object handshape (e.g., B-handshape, H-handshape, or 1-handshape which all resemble a saw). To illustrate, Adult Homesigner 1 and his brother used the same general handshape type (i.e., Handling, Object, Handling+Object) for 19 items, but only used the exact same specific handshape for 7 of those items (37%). In contrast, two unrelated hearing non-signers also used the same iconic handshape type for 19 items but used the same exact handshape for 12 of those items (63%). Further analysis will investigate whether the reported conventionalization of iconic handshape preferences is related to conventionalization of specific handshapes.”
- I found the first sentence vague "it is unclear where the conventionalization of iconic handshape preferences originates". What does this mean, where it originates? Isn't conventionalization part of all language change?
We changed the wording to be more clear: “it is unclear how iconic handshape preferences arise and become conventionalized.”
Reviewer 2 Report
This is a well written paper! This data set enables interesting handshape comparisons to investigate conventionalization within family homesign systems.
Two things came to mind that might enhance the paper. I believe Jill Morford did a comparison of the famous SGM homesigner David to his own sister (e.g., communication partner); I believe he even corrects her performance on the VMP (Supalla et al) test, suggesting that he has some sense of standards of form in his own homesign system. That might be a relevant study to situate your findings. Also, Laura Horton has used a statistical approach called the Jaccard Index to look for similarity or diversity between data sets (she also looks for conventionalization between groups of signers within family homesign systems) that might be useful for the repeated pairwise comparisons this study employs (I think it's in press now, contact her).
On p, 6 you talk about the 8 hearing nonsigning participants, describing that they have no experience with sign language or the homesigners. But the Table 2 describes the 8 as 4 being in "same family" and 4 "unrelated". At first I thought the 4 "same family" meant that they were in the same family as a homesigner. I think it means they all have no sign experience, but 4 of them happen to be within the same family. Maybe just re-work the word choice to make it clear. Also, Would you expect the same-family nonsigners to create gestures similarly? I also think it would be worth spending a bit more discussion about what you think the hearing nonsigners are doing on this task and what you predict about their performance (It's interesting that the hearing unrelated have 70% conventionalization...what explains that?)
Minor typos:
Several times the word author is typed as auhtor.
In Table 1 in the row about lexical frequency, it should be word, not work, right?
In Figure 1 caption, I think there is scrambled wording and part (c) is not described.
Page 13, line 500: seems to be an errant line break
Author Response
- I believe Jill Morford did a comparison of the famous SGM homesigner David to his own sister (e.g., communication partner); I believe he even corrects her performance on the VMP (Supalla et al) test, suggesting that he has some sense of standards of form in his own homesign system. That might be a relevant study to situate your findings.
We cited and discussed the findings of this study on page 14:
“Previous work by Singleton, Morford and Goldin-Meadow (1993) comparing the prodictions of the homesigner called David with those of his hearing sister showed that her gestures more closely resembled those of non-signers rather than those of her homesigning brother. Frequent and prolonged interaction between communication partners and homesigners does not appear to be enough to conventionalize gestures in a homesign system. As David was documented correcting his sister’s gesture forms, so has [author] observed adult homesigners correcting their family members, providing similar evidence of standards of form in homesign systems that communication partners do not always pick up on.”
- Laura Horton has used a statistical approach called the Jaccard Index to look for similarity or diversity between data sets (she also looks for conventionalization between groups of signers within family homesign systems) that might be useful for the repeated pairwise comparisons this study employs (I think it's in press now, contact her).
We originally discussed using the Jaccard Index but ultimately decided not to because it only provides information about similarity with respect to the homesigner and we wanted to assess the degree of conventionalization across the entire family. Therefore we used repeated pairwise comparisons in order to obtain overall family similarity instead of just the degree of similarity of each communication partner with the respective homesigner.
- On p, 6 you talk about the 8 hearing nonsigning participants, describing that they have no experience with sign language or the homesigners. But the Table 2 describes the 8 as 4 being in "same family" and 4 "unrelated". At first I thought the 4 "same family" meant that they were in the same family as a homesigner. I think it means they all have no sign experience, but 4 of them happen to be within the same family. Maybe just re-work the word choice to make it clear.
In order to clarify any confusion, we added the sentence: “Note that the hearing non-signers who were from the same family did not have a homesigner in their family; further, none of the unrelated hearing non-signers had a homesigner in their families.” We also reworded Table 2 so that the hearing non-signers are referred to as “related” and “unrelated”.
- Would you expect the same-family nonsigners to create gestures similarly? I also think it would be worth spending a bit more discussion about what you think the hearing nonsigners are doing on this task and what you predict about their performance (It's interesting that the hearing unrelated have 70% conventionalization...what explains that?)
We elaborated on these questions more in the discussion on page 14:
“We included the hearing non-signing family and the group of unrelated hearing non-signers in order to pull apart the influence of having a homesigner in a family versus generally interacting within a family; however, it was somewhat surprising that the hearing unrelated non-signers displayed such high conventionalization. Note that our measure of conventionalization does not take into account the complexity of the handshapes produced. Thus, the hearing people in our study may be producing the most straightforward and least complex handshape forms. In their context as hearing people who do not regularly communicate with a deaf person, they are not burdened with additional iconicity demands (e.g., using patterned iconicity systematically) and may therefore consider fewer, and less complex, handshape options. Since hearing non-signing gesturers already tend to both produce less finger and joint complexity than homesigners (author; author) and prefer Handling handshapes (Padden et al., 2015), this may partly be responsible for the convergence. In other words, the handling handshape preference may also be generated independently in each individual, rather than due to convergence in a group. In addition, it follows that they would exhibit restricted options for types of handshapes, and their productions therefore may appear more conventionalized, even if little actual conventionalization has occurred, with the apparent similarity merely reflecting similar strategies. In other words, although the hearing unrelated non-signers may appear to be highly conventionalized, they are likely using the simplest forms at their disposal, and given their limited handshape options, just happen to be using the same simple handshape types.”
- Minor typos
All minor typos have been corrected.
Reviewer 3 Report
This is an important, well-designed study and a well-written paper. I have only one minor recommendation: Please indicate the total number of male and female participants (lines 249 and 255), as we can’t assume a gender binary.
Author Response
Both female and male participant numbers are now included in the participants section.
Reviewer 4 Report
See attached file

Author Response
General observation (concerning the Introduction): I find a bit surprising that the question of sensible periods for language acquisition is not even mentioned. I see that the results of this paper do not offer any firm evidence to advance on this issue, still it is kind of strange that it is totally ignored in a paper in which homesigners’ production is studied. On this see my last observation below on some surprising findings concerning (lack of) conventionalization.
We originally did not include a discussion on the sensitive period for language acquisition because it is out of the scope of this paper. The focus of the current paper is how conventionalization of handshapes occurs in families with a homesigner. It is important to note that within these families, the deaf homesigner does not have access to the spoken language used by the family, and no already-existing sign language is used by anyone in the family. Thus, the family is collaboratively innovating a communication system, though this collaboration is asymmetric. Therefore, a discussion of the sensitive period for language acquisition feels outside the scope of the article. However, the question of a sensitive period as it relates to the hearing communication partners acquiring homesign systems from the deaf homesigner in their family is addressed systematically in Carrigan & Coppola 2017, which found that the younger a communication partner was when they first began using the homesign system, the higher they scored on a test of comprehension of homesign utterances (https://pubmed.ncbi.nlm.nih.gov/27771538/).
In light of the reviewer’s comments we have brought up the idea of sensitive periods for language acquisition briefly with regard specifically to communication partner’s age and homesign experience analyses:
(lines 209-213) “This result is consistent with the idea that homesign systems are sufficiently similar to languages with longer histories and more developed structure that they also show hallmarks of a sensitive period for acquiring them among those who are exposed to them at different ages (Mayberry & Kluender, 2018; Newport, Bavelier & Neville, 2001).”
(lines 242-245) “This is also related to the sensitive period for language acquisition which research demonstrates a relationship between age of exposure to a language and proficiency in that language (e.g., Newport, 1990; Mayberry & Fischer, 1989; Emmorey & Corina, 1990).”
Line 67: The sentence ‘For nouns, the focus of the current study, handshape type is often more uniform’ is somewhat unclear. I think that Author means that handshape type is uniform within a given language, not across languages. If so, please clarify.
“Within a given language” was included to clarify.
Line 90: the concept of ‘patterned iconicity’ is not defined (only a reference to Padden’s work is given). I think a brief definition is needed to make this paper more self-contained.
Patterned iconicity is now more clearly defined in the first paragraph of the introduction (lines 35-37): “​​Research on “patterned iconicity”, a term created by Padden and her colleagues which refers to repeated use of iconic strategies for signs within a certain category (Padden et al. 2013, 2015; Hwang et al., 2017), has demonstrated shared preferences for different types of iconicity ​​when naming objects in both sign languages and in the gestures made by hearing people. ”
Line 226: when I read the presentation of the research, my initial reaction was: why running two separate studies? I take the point that Study 1 targets factors related to the participants (say, their age or their familiarity with a homesign system) and how this may influence handshape preferences, while Study 2 targets factors related to the experimental items (say,the types of object to be described) that also modulate the handshape preferences. But: why not combining the two studies, for example by investigating how familiarity with a homesign system interacts with the type of stimuli (say, tools as opposed to non-tools or familiar as opposed to less familiar)? Was this due to the difficult of running this type of analyses with a limited number of data point? Please clarify.
Correct, we did not have enough data to run more detailed analyses such as the ones proposed above. While these are very interesting questions, we are simply unable to do so with the current dataset.
Line 284: What lead to the decision of not randomizing the slides? Did the Author check whether there was a trend (say preference for Object and the beginning or Handling at the end of the slide presentation or the other way around)?
First, we used the stimuli with permission from Padden and colleagues, and followed the same protocol that they did. This helps make the data we collected more comparable to data collected in other languages/contexts, which is a high priority given the rarity of empirical data collected in emerging language situations. Second, data collection in Nicaragua is difficult and unexpected things happen regularly; thus, keeping the same stimuli order made things easier for those collecting data and for those coding the data a decade later. We did not find any order effects or any differences between early and later presented stimuli. On average, the first half of the items and the second half both had 55% Handling and 35-36% Object handshapes.
Line 293: There is no picture illustrating the Handling-Object-Simultaneous option. It would be nice to add a picture to this effect.
We don’t really have a clear example of H+O SIM that translates well to a still image. We have now included a video example of H+O SIM in the supplementary materials.
Line 290, Frame (b) in Figure 1. What is non-dominant hand doing? Is it representing a part of the saw as this object was depicted in the slide? Without seeing the original slide, it is hard to decide whether frame 2 contains a Handling Handshape or an Object Handshape. Please add the original slide here.
As stated in the materials and procedure section, the stimuli slides were still images of the different tools/objects (please see Padden et al., 2013; 2015 for examples of the stimuli slides). Sometimes, participants would extrapolate from the images when labeling the tool so even if there was just an image of a saw, they would include extra details in their descriptions. In Fig 1b, the non-dominant hand is likely representing an object that a saw might cut. We only included in the analysis handshapes that were specifically in reference to the tool (lines 315-316 “Signs marked as Other (i.e., not specifically in reference to the tool or not iconic) were not included in the analysis”) so in this example, the non-dominant hand is not representing the tool and therefore is not relevant to the analysis.
Line 321: The sentence “Do homesigners and communication partners show a universal handshape preference for iconicity (Handling/Object)?” can be read as question about the presence of an iconic strategy, as opposed to a non-iconic one (this is a classical question in sign language studies). However, I imagine that the research question here is different, namely: since they have to adopt an iconic structure, do participants express iconicity by choosing a Handling handshape or an Object handshape? If I understand correctly, please clarify.
We changed the wording of our research question to be more clear: “Do homesigners and communication partners tend to express iconicity by using a Handling handshape or an Object handshape?”
Line 405: “... higher conventionalization was associated with being younger when tested, and with fewer years of experience using a homesign system. A weak inverse correlation was found between conventionalization and age of first exposure (rs = -0.22, p > 0.05), showing that higher conventionalization was somewhat associated with being exposed to a homesign system from a young age.”
Although there is no contradiction between these two findings, there is some tension between them, as prima facie they suggest that exposition to the homesign system induces conventionalization in a case but hinders it in another case. I suggest that Author points out this tension and refer to the General Discussion where the issue id further discussed. In fact, in the general discussion (around Line 528), the Author proposes an explanation for the first finding that goes as follows: young people and people unexperienced with homesigns use a more restricted inventory of handshapes. Although this may be misinterpreted as presence of more conventionalization, in fact no conventionalization takes place to begin with, as by definition a limited choice leads to less variability (even in absence of a conventionalization process). I find this explanation interesting, also because it does not extend to the type of conventionalization associated with being exposed to a homesign system from a young age (if anything, exposition from a young age should allow the diversification of handshapes if something like a sensitive period applies to acquisition of homesigns).
We included this sentence directly after the above excerpt (lines 433-434): “While these findings do not necessarily contradict one another, they do raise some questions which are considered in the discussion section.”